# 3D Object Detection for Self-Driving Cars Using Video and LiDAR: An Ablation Study

**DOI:** 10.3390/s23063223

**Published:** 2023-03-17

**Authors:** Pascal Housam Salmane, Josué Manuel Rivera Velázquez, Louahdi Khoudour, Nguyen Anh Minh Mai, Pierre Duthon, Alain Crouzil, Guillaume Saint Pierre, Sergio A. Velastin

**Affiliations:** 1Cerema Occitanie, Research Team “Intelligent Transport Systems”, 1 Avenue du Colonel Roche, 31400 Toulouse, France; 2Cerema Centre-Est, Research Team “Intelligent Transport Systems”, 8-10, Rue Bernard Palissy, 63017 Clermont-Ferrand, France; 3Institut de Recherche en Informatique de Toulouse IRIT, University of Toulouse, UPS, 31062 Toulouse, France; 4Department of Computer Science and Engineering, Universidad Carlos III de Madrid, Leganés, 28911 Madrid, Spain; 5School of Electronic Engineering and Computer Science, Queen Mary University of London, London E1 4NS, UK

**Keywords:** autonomous vehicle, 3D object detection, LiDAR, fusion, stereo camera

## Abstract

Methods based on 64-beam LiDAR can provide very precise 3D object detection. However, highly accurate LiDAR sensors are extremely costly: a 64-beam model can cost approximately USD 75,000. We previously proposed SLS–Fusion (sparse LiDAR and stereo fusion) to fuse low-cost four-beam LiDAR with stereo cameras that outperform most advanced stereo–LiDAR fusion methods. In this paper, and according to the number of LiDAR beams used, we analyzed how the stereo and LiDAR sensors contributed to the performance of the SLS–Fusion model for 3D object detection. Data coming from the stereo camera play a significant role in the fusion model. However, it is necessary to quantify this contribution and identify the variations in such a contribution with respect to the number of LiDAR beams used inside the model. Thus, to evaluate the roles of the parts of the SLS–Fusion network that represent LiDAR and stereo camera architectures, we propose dividing the model into two independent decoder networks. The results of this study show that—starting from four beams—increasing the number of LiDAR beams has no significant impact on the SLS–Fusion performance. The presented results can guide the design decisions by practitioners.

## 1. Introduction

Object detection is one of the main components of computer vision aimed at detecting and classifying objects in digital images. Although there is great interest in the subject of 2D object detection, the scope of detection tools has increased with the introduction of 3D object detection, which has become an extremely popular topic, especially for autonomous driving. In this case, 3D object detection is more relevant than 2D object detection since it provides more spatial information: location, direction, and size.

For each object of interest in an image, a 3D object detector produces a 3D bounding box with its corresponding class label. A 3D bounding box can be encoded as a set of seven parameters [1]: (x,y,z,h,w,l,θ), including the coordinates of the object center (x,y,z), the size of the object (height, width, and length), and its heading angle (θ). At the hardware level, the technology involved in the object detection process mainly includes the use of mono and stereo cameras, with visible light or infrared cameras, RADAR (radio detection and ranging), and LiDAR (light detection and ranging), and gated cameras. In fact, the current top-performing methods in 3D object detection are based on the use of LiDAR (Figure 1) [2,3].

However, highly accurate LiDAR sensors are extremely costly (the price of a 64-beam model is around USD 75,000 [4]), which incurs a hefty premium for autonomous driving hardware. Alternatively, systems based only on camera sensors have also received much attention because of their low costs and wide range of use. For example, in [5], the authors claim that instead of using expensive LiDAR sensors for accurate depth information, the alternative is to use pseudo-LiDAR, which has been introduced as a promising alternative at a much lower cost based solely on stereo images. The paper presents the advances to the pseudo-LiDAR framework through improvements in stereo depth estimation. Similarly, in [6], instead of using a LiDAR sensor, the authors provide a simple and effective one-stage stereo-based 3D detection pipeline that jointly estimates depth and detects 3D objects in an end-to-end learning manner. These authors claim that this method outperforms previous stereo-based 3D detectors and even achieves comparable performance to a few LiDAR-based methods on the KITTI 3D object detection leaderboard. Another example is presented in [7]. To tackle the problem of high variance in depth estimation accuracy with a video sensor, the authors propose CG-Stereo, a confidence-guided stereo 3D object detection pipeline that uses separate decoders for foreground and background pixels during depth estimation, and leverages the confidence estimation from the depth estimation network as a soft attention mechanism in the 3D object detector. The authors say that their approach outperforms all state-of-the-art stereo-based 3D detectors on the KITTI benchmark.

Another interesting solution presented in the literature is the combination of LiDAR and a stereo camera. These methods exploit the fact that LiDAR will complete the vision and information provided by the camera, by adding notions of size and distance to the different objects that make up the environment. For example, the proposed method in [8] takes advantage of the fact that it is possible to reconstruct a 3D environment using images from stereo cameras, making it possible to extract a depth map from stereo camera information and enrich it with the data provided by the LiDAR sensor (height, width, length, and heading angle).

In [9], a new method proposed by us, called SLS–Fusion (sparse LiDAR and stereo fusion network), is presented. This is an architecture based on DeepLiDAR [10] as a backbone network and the pseudo-LiDAR pipeline [8] to fuse information coming from a four-beam LiDAR and a stereo camera via a neural network. Fusion was carried out to improve depth estimation, resulting in better dense depth maps and, thereby, improving 3D object detection performance. This architecture is extremely attractive in terms of cost-effectiveness, since 4-beam LiDAR is much cheaper than 64-beam LiDAR (the price of a 4-beam model is around USD 600 [11]). Results presented in Table 1 and in [9] show that the performance offered by the 64-beam LiDAR (results were obtained with PointRCNN [12] from testing on the KITTI dataset [13], for “Car” objects with IoU = 0.5 on three levels of difficulty (defined in [14]): easy (fully visible, max. truncation–15%) = 97.3, moderate (partly occluded, max. truncation–30%) = 89.9, hard (difficult to see, max. truncation–50%) = 89.4) is not far from the one reached by the stereo camera and the four-beam LiDAR model (with the best results obtained from testing on the KITTI dataset, for “Car” objects with IoU = 0.5 on three levels of difficulty: easy = 93.16 and moderate = 88.81 with SLS–Fusion, and hard = 84.6 with Pseudo-LiDAR++). For this comparison, the satisfactory results obtained by the 64-beam LiDAR were modeled directly in PointRCNN, while the combination of video and LiDAR requires the generation of a new point cloud, usually referred to as the pseudo-point cloud.

There are other solutions presented in the literature based only on stereo cameras, such as the CG-Stereo method presented in [7], which achieves outstanding results on easy mode (level of difficulty: easy = 97.04, see Table 1). However, implementing two sensors (e.g., LiDAR and stereo camera) instead of one brings robustness to the 3D object detection system, as demonstrated in [15]. In addition, the SLS–Fusion and Pseudo-LiDAR++ methods show better results in the hard mode, as illustrated in Table 1.

**Table 1 sensors-23-03223-t001:** Evaluation of the 3D object detection part of SLS–Fusion compared to other competitive methods. Average precision $AP_{BEV}$ [16] results on the KITTI validation set [13] for the “Car” category with the IoU at 0.5 and on three levels of difficulty, (defined as [14]): easy, moderate, and hard. S, L4, and L64, respectively, denote stereo, simulated 4-beam LiDAR, and 64-beam LiDAR. According to the inputs, the maximum average precision values are highlighted in bold.

Method	Input	Easy	Moderate	Hard
TLNet [17]	S	62.46	45.99	41.92
Stereo-RCNN [18]	S	87.13	74.11	58.93
Pseudo-LiDAR [8]	S	88.40	76.60	69.00
CG-Stereo [7]	S	**97.04**	**88.58**	**80.34**
Pseudo-LiDAR++ [5]	S+L4	90.30	87.70	**84.60**
SLS–Fusion [9]	S+L4	**93.16**	**88.81**	83.35
PointRCNN [12]	L64	**97.30**	**89.90**	**89.40**

From the above, the interest in using a low-cost LiDAR and stereo camera model as an alternative solution is justifiable. However, there is still a need to understand the scope and limitations of a 3D object detection model composed of LiDAR and a stereo camera. Knowing exactly what is the role of each sensor in the performance of the architecture will allow optimization of the synergy of these two sensors, possibly reaching higher accuracy levels at lower costs.

In this study, after analyzing the fusion between the stereo camera and LiDAR for 3D object detection, we studied the respective role of each sensor involved in the fusion process. In particular, an ablation study was conducted considering LiDAR with different beam number for object detection. Thus, LiDAR with 4, 8, 16, and 64 beams, either alone or fused with the stereo camera were tested. Regardless of the number of beams, fusion with stereo video always brought the best results. On the other hand, to reduce the overall equipment costs, the fusion between a 4-beam LiDAR and a stereo camera was enough to obtain acceptable results. Thus, when merging LiDAR with video, it is not necessary to use a LiDAR with a higher number of beams (which is more expensive).

Thus, a detailed study of the relationship between the number of beams of a LiDAR and the accuracy obtained in 3D object detection using the SLS–Fusion architecture is presented here. Roughly, two important results are presented: (1) an analysis of the camera stereo and LiDAR contributions in the performance of the SLS–Fusion model for 3D object detection; and (2) an analysis of the relationship between the number of LiDAR beams and the accuracy achieved by the 3D object detector. Both analyses were carried out by an ablation method [19], which was carried out by removing one component from the architecture to understand how the other components of the system performed. This characterizes the impact of every action on the overall performance and ability of the system.

After this introduction, to make the paper more self-contained, Section 2 presents the work related to the sensor LiDAR fusion technique. Section 3 describes the framework by detailing the main contributions. Section 4 explains the main characteristics of combining a stereo camera and LiDAR in the SLS–Fusion architecture. Section 5 evaluates the contribution of each component to the neural network fusion architecture. Finally, Section 6 presents the concluding remarks, lessons learned, and some advice for practitioners.

## 2. Related Work

The automation of driving is based on many aspects, such as perception, positioning, scenario analysis, decision-making, and command control. The perception sub-system is one of the most critical elements, since the behavior of the autonomous vehicle (AV) depends entirely on the objects that are around it. Furthermore, the safety of the users who share the road with the AV depends on them being detected and identified by the AV. Consequently, object detection has become one of the most critical research areas for the perception of self-driving cars.

Currently, vision systems combine visible imaging, LiDAR, and/or RADAR technology to perceive the vehicle’s surroundings. Moreover, thermal cameras have been introduced to respond to a lack of visibility in low light conditions or adverse weather conditions [20]. Despite this, LiDAR and stereo cameras continue to be the most widely used sensors for vehicle environment perception. This is mainly due to the wealth of information offered by these two sensors:Visible light cameras are passive sensors used as the “eyes” of automated driving vehicles. Unlike thermal cameras, stereo cameras provide information about the textures and colors of objects in the image, which leads to better results when they are used for the detection of other road users, road signs, traffic lights, and lane markings. In addition, stereo camera systems use inherent 3D capabilities.LiDAR involves active devices that operate by emitting pulsed light laser and measuring its reflection time. From the measurements, a 3D map of the environment can be generated. In addition, LiDAR can provide higher angular resolution than radars (due to their shorter wavelengths), resulting in higher accuracy when implemented for edge identification.

Part of the interest in these two sensors is because both are capable of providing information necessary for 3D object detection. However, the estimation of contours, shapes, and textures is better in the stereo camera, while LiDAR provides higher accuracy for depth estimation. To obtain better results, data fusion is usually performed to exploit the advantages of both sensing systems.

### 2.1. Camera-LiDAR-Based 3D Object Detection Methods

Three-dimensional (3D) object detection is a topic that has gained interest within the scientific community dedicated to vehicle automation. Based on LiDAR and stereo cameras, and considering only deep learning-based approaches, 3D object detection methods are classified according to the type of input data: camera-based, LiDAR-based, and fusion-based 3D object detection.

#### 2.1.1. Camera-Based Methods

Some of the first algorithms were keypoint/shape-based methods. In 2017, Chabot et al. [21] presented Deep MANTA, one of the first works on camera-based 3D object detection. This architecture recognizes 2D keypoints and uses 2D-to-3D matching. This algorithm has two steps: (1) a RCNN architecture to detect and refine 2D bounding boxes and keypoints, and (2) a predicted template similarity to pick the best matching 3D model inside a 3D dataset. However, the main disadvantage of this method is the excessive time required for 2D/3D matching.

On the other hand, the pseudo-point cloud-based methods arose from the idea of simulating LiDAR from a stereo camera. These methods typically convert data from 2D to 3D by using extrinsic calibration information between the camera and LiDAR. For example, in Xu and Chen [22], a depth map from RGB images was predicted and then concatenated as RGB-D (where D is the datum provided by the depth map) to form a tensor of six channels (RGB image, Z-depth, height, distance) used to regress the 3D bounding boxes. For example, the Pseudo-LiDAR method presented by Wang et al. [8] was inspired by that article, based on the idea of data representation transformation to estimate the depth map from an RGB image using a depth estimation neural network. Next, the predicted depth map is converted into a pseudo-3D point cloud by projecting all pixels with depth information into LiDAR coordinates. Then, the pseudo-point cloud was ready to be used as input in any LiDAR-based object detection method. The ability to reconstruct a 3D point cloud from less expensive monocular or stereo cameras is a valuable feature of this approach. Based on Pseudo-LiDAR [8], the same authors proposed Pseudo-LiDAR++ [5] through a stereo depth estimation neural network (SDN). They proposed a depth cost volume to directly predict the depth map instead of predicting the disparity map, as proposed by Chang and Chen [23]. This boosts the predicted depth map accuracy. In addition, to improve the accuracy of the predicted depth map, they proposed a depth correction phase using a simulated four-beam LiDAR to regularise the predicted depth map.

#### 2.1.2. LiDAR-Based Methods

Due to the irregular, unstructured, and unordered nature of point clouds [24], they are often handled in one of three ways: projecting point clouds to generate a regular pseudo-image, sub-sampling point cloud cells called voxels, or encoding raw point clouds with a sequence of multi-layer perceptron (MLP), as proposed in [25]. For 3D object detection, LiDAR-based methods are usually classified into four categories: view-based, voxel-based, point-based, and hybrid point-voxel-based detection.

View-based detection methods, such as the one presented by Beltran et al. [26], project a point cloud onto a 2D image space to obtain a regular structure as an initial stage. Generally, a CNN (convolutional neural network) is then used to take advantage of this information [27,28]. The most common types of projection are bird’s eye view (BEV), front view (FV) [16], range view (RV) [29], and spherical view (SV) [30].

The voxel-based method maps a point cloud into 3D grids (voxels) as an initial stage. In 2017, Engelcke et al. [31] presented their LiDAR-based detector Vote3Deep. In this method, LiDAR point clouds are discretized into a sparse 3D grid. Then, items are detected using a sliding-window search with a fixed-size window with *N* different orientations. In each window, a CNN performs binary classification.

Point-based methods usually deal with the raw point cloud directly instead of converting the point cloud to a regular structure. Qi et al. [25] introduced PointNet, a pioneering study on deep learning-based architectures for processing point cloud raw data for classification and semantic segmentation. They argue that as the point cloud is unordered, the architecture should be permutation-invariant for all points.

#### 2.1.3. Fusion-Based Methods

Image-based object detection is an advanced research area. In addition, cameras are cheap and provide a lot of texture information about objects based on color and edges. However, images lack depth information, which is extremely important for 3D tasks. Even with a stereo camera, the resulting depth map lacks accuracy. On the other hand, although LiDAR does not give texture information, LiDAR-based methods show a very high performance compared to camera-based methods. However, there are still limitations (such as obscure information) regarding object categories. For example, in some cases, it is difficult to distinguish whether it is a car or a bush based on point cloud data alone, while this can be handled more easily by looking at the image data. This is why methods based on data fusion have been developed exploiting the advantages of both sensors. In the literature, there are three main fusion methods: early fusion, where the raw data are fused at the data level or feature level to form a tensor data of numerous channels; late fusion, where the fusion takes place at the decision level; and deep fusion, where fusion is carefully constructed to combine the advantages of both early and late fusion systems.

For example, Qi et al. [1] presented the Frustum-PointNet architecture, which is composed of three phases: 3D frustum proposal, 3D instance segmentation, and 3D bounding box estimation. The first phase of this procedure is to produce 2D region proposals. By extruding the matching 2D region proposal under a 3D projection, a 3D frustum proposal is generated. The instance segmentation stage feeds the frustum proposal point cloud to the PointNet segmentation network [25], which classifies each point and determines if it is linked with the discovered item. In the last stage, all positively classified points are loaded into a new PointNet that estimates 3D bounding box parameters.

Chen et al. [16] introduced MV3D, where the LiDAR point cloud is projected onto both a 2D top view and a 2D front view, from which feature maps are extracted using two separate CNN. The LiDAR top-view feature map is passed to an RPN (Region Proposal Network) to output proposal 3D bounding boxes. Each 3D proposal is projected onto the feature maps of all three views and a fixed-size feature vector is extracted for each view using pooling. The three feature vectors are then fused in a region-based fusion network, which finally outputs class scores and regresses 3D bounding box residuals.

A similar approach, also utilizing the PointNet architecture, was independently presented in Xu et al. [32]. Just as in Frustum-PointNet, a 2D object detector is used to extract 2D region proposals (ResNet), which are extruded to the corresponding frustum point cloud. Each frustum is fed to a PointNet, extracting both point-wise feature vectors and a global LiDAR feature vector. Each 2D image region is also fed to a CNN that extracts an image feature vector. For each point in the frustum, its point-wise feature vector is concatenated with both the global LiDAR feature vector and the image feature vector. This concatenated vector is finally fed to a shared MLP, outputting 8 × 3 values for each point. The output corresponds to predicted (x,y,z) offsets relative to the point for each of the eight 3D bounding box corners. The points in the frustum are thus used as dense spatial anchors. The MLP also outputs a confidence score for each point, and in inference, the bounding box corresponding to the highest-scoring point is chosen as the final prediction.

Ku et al. [33] introduced another fusion architecture named AVOD. Here, the LiDAR point cloud is projected onto a 2D top-view, from which a feature map is extracted by a CNN. A second CNN is used to extract a feature map also from the input image. The two feature maps are shared by two subnetworks: an RPN and a second-stage detection network. The reported 3D detection performance is a slight improvement compared to [16]; it is comparable to that of [34] for cars but somewhat lower for pedestrians and cyclists. The authors also found that using both image and LiDAR features in the RPN, as compared to only using LiDAR features, has virtually no effect on the performance of cars but a significant positive effect for pedestrians and cyclists.

Similar to some of the methods mentioned above, the SLS–Fusion method presented in Mai et al. [9] resulted from this idea. Roughly, the SLS–Fusion method estimates the depth maps from a stereo camera and the projected LiDAR depth maps. However, as Zhu et al. [35] point out, this produces a mismatch between the resolution of point clouds and RGB images. Specifically, taking the sparse points as the multi-modal data aggregation locations causes severe information loss for high-resolution images, which in turn undermines the effectiveness of multi-sensor fusion.

More research is needed on the limits of a 3D object detection model composed of a LiDAR and a stereo camera. Knowing the role of each sensor will allow for the optimization and configuration of the proposed methods. With this in mind, this paper presents a detailed study of the relationship between the number of LiDAR beams and the accuracy obtained in 3D object detection using the SLS–Fusion architecture.

## 3. Analysis of the Role of Each Sensor in the 3D Object Detection Task

SLS–Fusion is a fusion method for LiDAR and stereo cameras based on a deep neural network for the detection of 3D objects (see Figure 2). Firstly, an encoder–decoder based on a ResNet network is designed to extract and fuse left/right features from stereo camera images and project the LiDAR depth map. Secondly, the decoder network constructs a left and right depth map of optimized features through a depth cost volume model to predict the corrected depth. After the expected dense depth map is obtained, a pseudo-point cloud is generated using calibrated cameras. Finally, a LiDAR-based method for detecting 3D objects (PointRCNN [12]) is applied to the predicted pseudo-point cloud.

Section 1 shows previous results of SLS–Fusion on the KITTI dataset, which uses the refined work of PoinRCNN to predict the 3D bounding boxes of detected objects. Experience with the KITTI benchmark and the low-cost four-beam LiDAR shows that the SLS–Fusion proposed by us outperforms most advanced methods as presented in Table 1. However, compared to the original PointRCNN detector that uses the expensive 64-beam LiDAR, the SLS–Fusion performance is lower. The superiority of the 64-beam LiDAR, used without fusing with stereo cameras, is expected because LiDARs with a high number of beams can provide very precise depth information, but highly accurate LiDAR sensors are extremely costly. In this case, the higher the number of LiDAR beams (i.e., the higher the number of point clouds generated), the higher the cost of the LiDAR sensor (from USD 1000 to 75,000). This paper, thus, analyzes how the stereo and LiDAR sensors contribute to the performance of the SLS–Fusion model for 3D object detection. In addition, the performance impact of the number of LiDAR beams used in the SLS–Fusion model was also studied. As shown in Figure 2, to separate the parts of the SLS–Fusion network that represents LiDAR and stereo camera architectures, it is only necessary to divide the model’s decoder into independent decoder networks. The decoder inside the SLS–Fusion model is the only component responsible for fusing features between LiDAR and stereo sensors.

Given a pair of images from a stereo camera and a point cloud from a LiDAR as input to detect 3D objects, the SLS–Fusion deep learning approach [9] has shown a high performance in the 3D object detection task. The analysis of this performance focuses on the contribution of the neural network component of each sensor (LiDAR or stereo) and of the type of LiDAR selected for the overall architecture of the system. In this work, LiDAR sensors are compared in terms of the number of beams and are grouped into 3 main types: low-cost (4 or 8 beams), medium-cost (16 beams), and high-cost (32 or 64 beams). This kind of study, particularly in artificial intelligence, is known as an ablation study [19,36], which is used to understand the contribution of each component in the system by removing it, analyzing the output changes, and comparing them against the output of the complete system. This characterizes the impact of every action on the overall performance.

This type of study has become the best practice for machine learning research [37,38], as it provides an overview of the relative contribution of individual architectures and components to model performance. It consists of several trials such as removing a layer from a neural network, removing a regularizer, removing or replacing a component from the model architecture, optimizing the network, and then observing how that affects the performance of the model. However, as machine learning architectures become deeper and the training data increase [39], there is an explosion in the number of different architectural combinations that must be assessed to understand their relative performances. Therefore, we define the notion of ablation for this study as follows:Consequences of varying the number of layers for the 4- and 64-beam LiDAR on the results of SLS–Fusion.Consequences of retraining SLS–Fusion by separating the parts of stereo cameras and LiDAR architectures.Analyzing and discussing the characteristics of the neural network architecture used.Applying some metrics with precision–recall curves (areas inside curves, F1-scores, etc.) to evaluate detection results achieved by the study.

## 4. Characteristics of the Neural Network Architecture Used

The main component of the SLS–Fusion neural network, used to fuse or separate LiDAR and stereo camera features (for an ablation study), is the encoder–decoder component (see Figure 2 and Figure 3). It is the main part of the SLS–Fusion network that aims to enrich the feature maps and, thus, lead to better-predicted depth maps from the stereo camera and the projected LiDAR images. To understand all of this, we outline how the encoder–decoder component works and how it will help to improve the precision of the system when using low-, medium-, or high-cost LiDAR.

As shown in Figure 3, both the stereo camera and LiDAR encoders are composed of a series of residual blocks from the neural network ResNet, followed by step-down convolution to reduce the feature resolution of the input. ResNet is a group of residual neural network blocks and each residual block is a stack of layers placed in such a way that the output of one layer is taken and added to another deeper layer within the block, as shown in Figure 4. The main advantage of ResNet is its ability to prevent the accuracy from saturating and degrading rapidly during the training of deeper neural networks (networks with more than 20 layers). This advantage helps in choosing a network to be as deep as needed for the problem at hand. What we needed in this case, was to extract as much detailed features as possible from sparse LiDAR data and high-resolution stereo images. This process considerably assisted the decoder network to fuse the extracted features well.

The network of the decoder consists of adding the functions of both LiDAR and stereo encoders, then up-projecting the result to progressively increase the resolution of the features and generate a dense depth map as a decoder output. Because the sparse input of LiDAR is heavily linked to the depth decoder output, features related to the LiDAR sensor should contribute more to the decoder than features related to the stereo sensor. However, as the add operation promotes features on both sides [40], the decoder is encouraged to learn more features related to stereo images in order to be consistent with the features related to the sparse depth from LiDAR. In this way, whatever the type and associated resolution of the selected LiDAR (low-, medium-, or high-cost types), the decoder network will correctly learn merged features. Consequently, the SLS–Fusion network always outperforms all types of LiDAR sensors in 3D object detection, as shown in the next section.

## 5. Assessment of the Different Network Architectures Implemented

To assess the operation of the SLS–Fusion system, the KITTI dataset [13,41], one of the most common dataset for autonomous driving is used to train the neural network for dense depth estimation, pseudo-point cloud generation, and 3D objection detection. It has 7481 training samples and 7518 testing samples for both stereo and LiDAR.

In this section, the results obtained with each component of the SLS–Fusion model (stereo camera, 4- and 64-beam LiDAR) are presented to understand the impact of each component on the ultimate detection performance of 3D objects and show how results are affected. To do this, a complete ablation study was performed by disabling each component as previously explained, or by changing the number of LiDAR model component beams. As shown in Figure 5, increasing the number of LiDAR beams will increase the number of points that represent the targets detected by the LiDAR. The aim of this illustration is to show the difficulty when dealing with LiDAR data processing. Depending on the environment, some areas are full of detected points, while others are empty. Consequently, the LiDAR contribution to the performance of the object detection model will be enhanced. However, as shown in Table 2, increasing the number of beams from 4 to 64 beams will significantly increase the cost of the LiDAR sensor. An optimized solution involves selecting the appropriate number of corresponding LiDAR beams, which can provide a desired performance level. For a more comprehensive survey of the LiDARs available on the market, the reader is referred to [42].

### 5.1. Metrics

The indicators that we used and recall here seem basic for specialists but it is important for us to recall them briefly because they are used in the analysis later. To better understand the detection process and the results achieved by this study, detection assessment measurements were used to quantify the performance of our detection algorithm in various situations. Among the popular measures for reporting results, there are basic concepts and evaluation criteria used for object detection [51] as follows:Confidence level: object detection model output score linked to the bounding of the object detected.Intersection over union IoU: the ratio of the area of overlap between the predicted bounding box and the ground truth bounding box to the area of union between the two boxes. The most common IoU thresholds used are 0.5 and 0.7.Basic measures: true positive (TP), true negative (TN), false positive (FP), and false negative (FN).Precision: the number of true positive predictions divided by the total number of positive predictions.Recall: the number of true positive predictions divided by the total number of ground truth objects.

Precision–recall curve: The precision–recall curve [52] is a good way to evaluate the performance of an object detector as the confidence is changed. In the case of 3D object detection, to make things clearer, we provide an example to better understand how the precision–recall curve is plotted. Considering the detections as seen in Figure 6, there are 6 images with 10 ground truth objects represented by the red bounding boxes and 21 detected bounding boxes shown in green. Each green bounding box must have a confidence level greater than 50% to be considered as a detected object and is identified by a letter (B1, B2, …, B21).

Table 3 shows the bounding boxes with their corresponding confidences. The last column identifies the detections as TP or FP. In this example, a TP is considered if the IoU is greater than or equal to 0.2, otherwise, it is a FP.

For some images, there is more than one detection overlapping a ground truth (see images 2, 3, 4, 5, 6 from Figure 6). In those cases, the predicted box with the highest IoU is considered a TP and all others as FPs (in image 2: B5 is a TP while B4 is a FP because the IoU between B5 and the ground truth is greater than the IoU between B4 and the ground truth).

The precision–recall curve is plotted by calculating the precision and recall values of the accumulated TP or FP detections. For this, first, we need to order the detections by their confidence levels, then we calculate the precision and recall for each accumulated detection as shown in Table 4 (note that for the recall computation, the denominator term is constant and equal to 10 since ground truth boxes are constant irrespective of detection).

### 5.2. Ablation Results

This section deals with the use of precision–recall curves to better understand the effect and the role of each component of SLS–Fusion on the entire model performance. It corresponds to the stereo component and the LiDAR component (changing the number of LiDAR beams from 4 to 64). To do this evaluation, we used the KITTI evaluation benchmark of 3D bounding boxes or 2D bounding boxes in BEV to compute precision–recall curves for detection, as explained in the previous section. The BEV for autonomous vehicles is a vision monitoring system that is used for better evaluation of obstacle detection. This system normally includes between four and six fisheye cameras mounted around the car to provide right, left, and front views of the car’s surroundings.

Figure 7 shows the precision–recall (P–R) curves obtained by taking into account, respectively, stereo cameras, 4-beam LiDAR, 8-beam LiDAR, 16-beam LiDAR, and 64-beam LiDAR. As shown in that figure, an object detector is considered good if its precision stays high as the recall increases, which means that only relevant objects are detected (0 false positives = high precision) when finding all ground truth objects (0 false negatives = high recall). On the other hand, a poor object detector is one that needs to increase the number of detected objects (increasing false positives = lower precision) in order to retrieve all ground truth objects (high recall). That is why the P–R curve usually starts with high precision values, decreasing as recall increases. Finally, detection results are divided into three levels of difficulty (easy, moderate, or hard) mainly depending on the dimension of the bounding box and the level of occlusion of the detected objects, especially for cars.

In summary, the P–R curve represents the trade-off between precision (positive predictive value) and recall (sensitivity) for a binary classification model. In object detection, a good P–R curve is close to the top-right corner of the graph, indicating high precision and recall. To provide a comprehensive evaluation of the performance of object detection models, represented by the shape of the P–R curves, a new metric called “minimal recall” is added to the graph. The minimal recall is defined exactly as the first recall value obtained when the P–R curve starts to drop sharply and the precision score is always higher or equal to 0.7 (this value is fixed experimentally). The best detector is then the detector that can achieve a high precision score (higher than 0.7) while the minimal recall score is closest to 1. Graphically, this means that a model that achieves a low level of detection will have a “minimal recall” that follows the left side of the graph, while a model that achieves a high level of detection will have a “minimal recall” that follows the right side of the graph.

Based on this idea, the P–R curves obtained for 2D objects in BEV are always better than those obtained for 3D objects. This is because the level of inaccuracy in detecting bounding boxes in 3D is always greater than in 2D. However, detecting the surrounding cars in the BEV projection view reduces the precision of estimating the distance of detected objects (cars) from the autonomous vehicle. P–R curves for the stereo camera show better results than four-beam LiDAR (BEV/3D minimal recall is 0.6/0.4 for stereo; BEV/3D minimal recall is 0.4/0.18 for LiDAR for the hard level of difficulty). However, fusing the two sensors (stereo camera and four-beam LiDAR) improves the detection performance (BEV/3D minimal recall is 0.63/0.42 in the hard level of difficulty). On the other hand, when the number of beams of LiDAR passes from a low-cost 4-beam LiDAR to a high-cost 64-beam LiDAR, the detector provides the best P–R curves (BEV/3D minimal recall is 0.65/0.45 in the hard level of difficulty).

Another way of comparing object detection performance is to compute the area under the curve (AUC) of the P–R. The AUC can also be interpreted as the approximated average precision (AP) for all recall values between 0 and 1. In practice, AP is obtained by interpolating through all *n* points in such a way that: (1)$$\mathrm{AP}=\sum_{i=0}^{n}\left(r_{i+1}-r_{i}\right)*\max_{\tilde{r}:r\geq r_{i+1}}\left(p\left(\tilde{r}\right)\right)$$
where $p(\tilde{r})$ is the measured precision at recall $\tilde{r}$.

The statistical properties of various methods to estimate AUC were investigated by [53], together with different approaches to constructing 95% of the confidence interval (CI). The CI represents the range within which 95% of the values from the P–R curve are distributed. Thus, this parameter corresponds to the dispersion around the AUC.

Hence, using the AUC for performance, and the asymmetric *logit* intervals presented in [53] for constructing the CI, Table 5 presents the 3D obstacle detection performance and the corresponding CI for an IoU of 0.7. Each cell of this table contains a pair of numbers (A/B) corresponding to the results obtained with the $AP_{BEV}$/$AP_{3D}$ metrics. In the upper side of the table, we consider the stereo camera and the different LiDAR sensors taken separately. When the sensors are taken separately, the stereo provides the following results: 82.38/68.08%, 65.42/50.81%, and 57.81/46.07% going from easy to hard. If we consider the LiDAR as taken separately, we can see that the 64-beam LiDAR provides the best results: 87.83/75.44%, 75.75/60.84%, and 69.07/55.95% going from easy to hard. Considering the progression of the detection as a function of the number of beams, an almost linear progression from 4 to 64 beams can be observed.

The bottom of Table 5 presents the performance and CI resulting from the fusion between the stereo camera and the different types of LiDAR. In Table 5, we immediately notice that, when compared with the stereo camera alone, there is improvement in the 3D object detection when the stereo camera is fused with the LiDAR with the lowest number of beams (four-beam); this is true for all levels of object detection difficulties (easy, moderate, and hard). In addition, we note that the stereo camera and 4-beam LiDAR combination provides slightly better results than those obtained with the 64-beam LiDAR in the easy and moderate modes. On the other hand, what is surprising is that the detection performance barely improved when the number of beams increased (less than a 1% difference between S+L4 and S+L64). Moreover, CI values from S+L4 and S+L64 (the cheapest and the most expensive combinations) are compared. From this comparison, there is an overlap between the CI of S+L4 (e.g., easy = [86.96, 88.04]) and the CI of S+L64 (easy = [87.52, 88.58]) for all levels of object detection difficulties, meaning that for the fusion between the stereo camera and LiDAR, there is no significance when moving from one architecture to another.

The obtained results could be related to the dataset processed. Thus, to deepen this analysis, other datasets other than KITTI must be used. This is perspective work. In any case, the best solution is obtained by fusing both sensors. This proves that each component of the SLS–Fusion architecture effectively contributes to the final performance of the model, and we cannot eliminate these components of the neural network architecture in all possible cases: low-cost, medium-cost, or high-cost LiDAR sensors.

## 6. Conclusions

In this work, we analyzed the contribution of a stereo camera and different versions of LiDAR (4 to 64 beams) to the performance of the SLS–Fusion model in detecting 3D obstacles, through an ablation study. Based on the ablation analysis and the different measurements used to evaluate our detection algorithm, it has been shown that sensors performed better when fused. The quantitative results showed that the detection performance drops moderately with each component disabled (stereo camera or LiDAR) or by modifying the number of LiDAR beams, and the full model works best. Moreover, this fusion approach was found to be very useful in detecting 3D objects in foggy weather conditions [54]

This analysis allowed us to identify several inherent characteristics of video and LiDAR. The camera’s resolution provides an undeniable and important advantage over LiDAR, as it captures information through pixels, which makes a significant difference even if the number of layers for the LiDAR is increased. Using two cameras makes it possible to measure distances to obstacles while keeping the same resolution because depth is calculated on all the pixels (dense stereo vision). LiDAR is mainly useful for determining distances. By extension, it also allows us to know the size and volume of objects very precisely, which can be extremely useful when classifying objects (cars, pedestrians, etc.).

In terms of resolution, LiDAR is limited to the fact that each of its pixels is a laser. A laser light means that we have a focused light. It is a point that does not deform and it allows high precision. It is more complicated to multiply the lasers in a very small space, and that is why, for the moment, LiDAR has a much lower resolution than the camera. A classic smartphone-type camera provides 8 million pixels per image, while LiDAR will have around 30,000 pixels (at the most). An advantage of LiDAR is its ability to adapt to changes in light, which is a strong disadvantage for imaging. As a consequence, the two types of sensors must be used in a complementary way.

In conclusion, SLS–Fusion is an effective obstacle detection solution for low and high cost LiDAR when combined with a stereo camera; an optimal cost-effective solution is achieved with the most economical four-beam LiDAR component. To better generalize the SLS-Fusion model and find the optimal balance between obstacle detection performance and the cost of the LiDAR component, it is desirable to test the model on various datasets and environments, such as waymo [55], nuScenes [56], or argoverse2 [57].

## 7. Perspectives to Go Further

To better understand the role and contribution of each technology to obstacle detection, it is necessary to make a more detailed analysis of the objects detected by one sensor or the other. Each type of sensor detects a list of objects with their 3D positions.

It is then necessary to merge the two lists of objects by following a rigorous procedure. In our system, when we consider the sensors separately, each provides a list of detected objects (included in 3D boxes) belonging to the same scene. We can develop a fusion module that would take as input the two lists of objects detected by the two types of sensors. In this case, the fusion module takes as input the list of the detected objects provided by both kinds of sensors and delivers a fused list of detected objects. For each object, we have the centroid of the bounding box, the class of the object, and the number of sensors that detected the object. To perform fusion between data from LiDAR and stereo vision objects, we could, for example, project the objects detected by stereo vision processing onto the laser plane.

**Object association:** In this step, we determine which stereo objects are to be associated with which LiDAR objects from the two object lists using, for example, the nearest neighbor technique. We could define a distance between the centroids of the objects detected by stereo and LiDAR. Then we can associate the current stereo object to the nearest LiDAR object from the stereo object, using as a reference point the coordinate points of the sensors installed on the vehicle. Exploiting the depths calculated by the stereo and the LiDAR, we only need to compare objects whose centroids are very close to each other (with a threshold) from the reference point. The result of this fusion process is a new list of fused objects. This list has the LiDAR objects, which could not be associated with stereo objects, and all of the stereo objects, which could not be associated with some LiDAR objects. By doing this, we can more objectively analyze the advantages and disadvantages of the two technologies, in what circumstance, for what type of object, at what distance, and with what brightness. The output of the fusion process consists of a fused list of objects. For each object, we have position (centroid) information, dynamic state information, classification information, and a count of the number of sensors (and for how many beams) detecting this object. This work is under development.

## Figures and Tables

**Figure 1 sensors-23-03223-f001:**
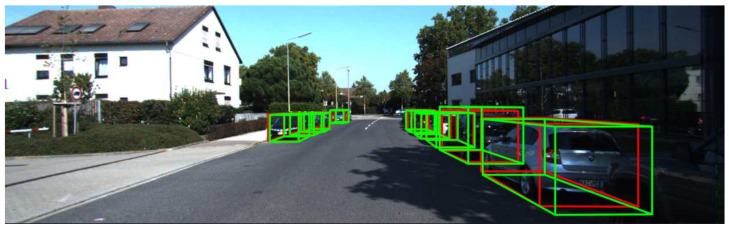
Example of a road scene where detection results using LiDAR technology is used. Detected objects are surrounded by bounding boxes. The green boxes represent detection while the red ones represent ground truth.

**Figure 2 sensors-23-03223-f002:**
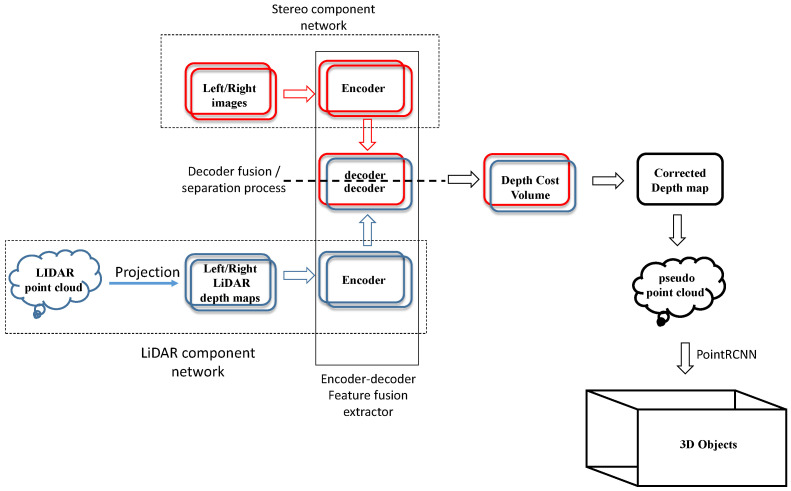
Overall structure of the SLS–Fusion neural network: red, blue, and red–blue boxes represent, respectively, stereo, LiDAR, and fusion networks: The LiDAR and stereo camera data are considered as inputs. Subsequently, in the encoder/decoder process, the resulting features are merged to obtain a depth map. Afterward, the depth map is converted into a point cloud, which makes it possible to estimate the depth of the objects detected by the two sensors.

**Figure 3 sensors-23-03223-f003:**
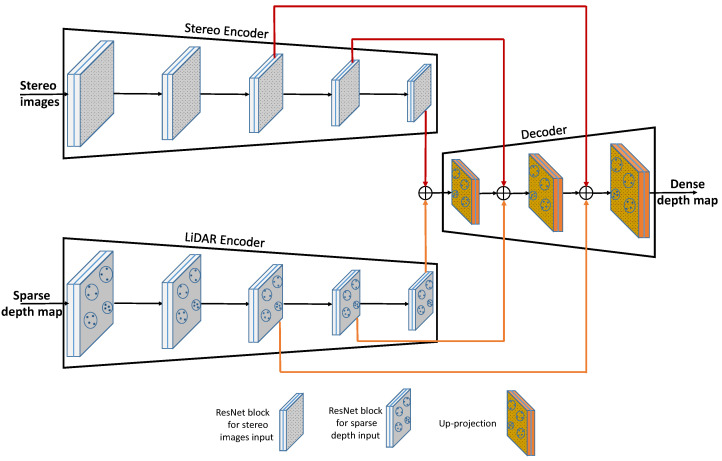
SLS–Fusion encoder–decoder architecture: The residual neural network blocks (ResNet blocks) within the encoder are used to extract features from the LiDAR and stereo inputs. The fusion process inside the decoder is accomplished through the use of addition and up-projection operators.

**Figure 4 sensors-23-03223-f004:**
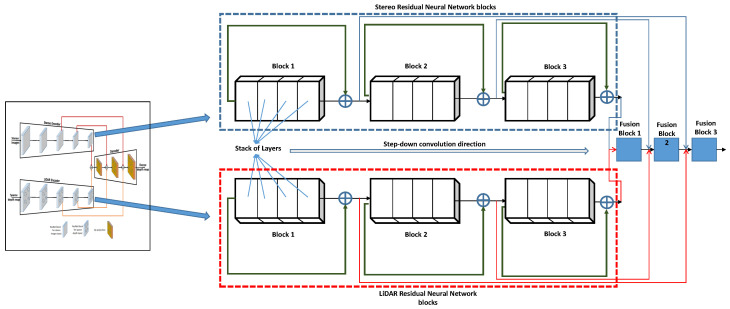
The structure of Stereo and LiDAR residual blocks inside the encoder/decoder of the SLS–Fusion model. A stack of layers is grouped into blocks for stereo and LiDAR networks, conducted by a step-down convolution direction and followed by a set of fusion blocks.

**Figure 5 sensors-23-03223-f005:**
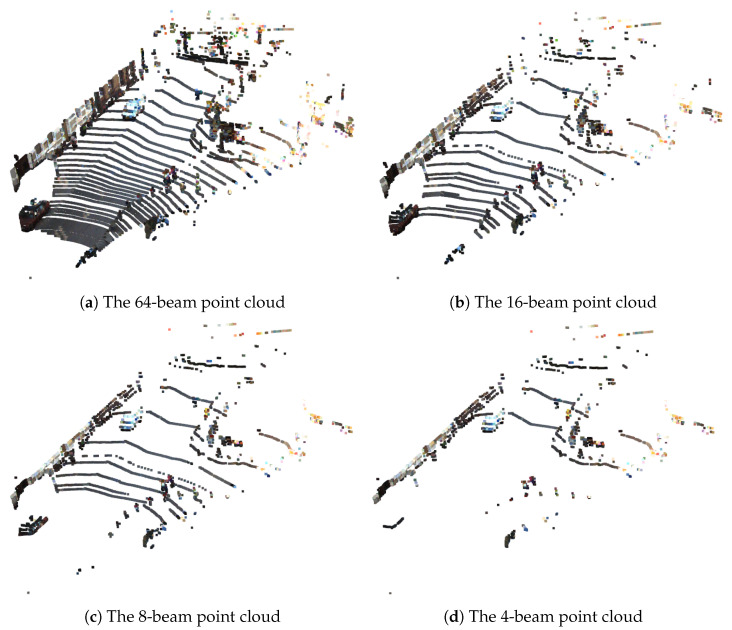
LiDARpoint clouds representing the measured environment. The point cloud is colored according to the information coming from the RGB image. The number of targets (points) varies according to the version of the LiDAR (number of beams): very dense for 64 beams (upper left) and dispersed for 4 beams (bottom right).

**Figure 6 sensors-23-03223-f006:**
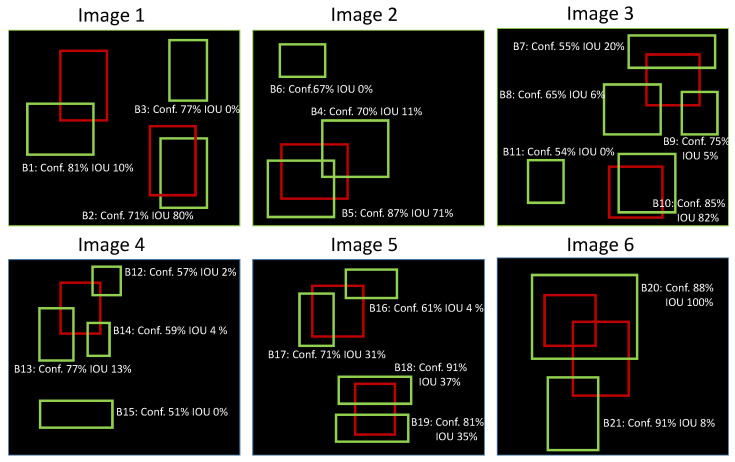
Example of how the precision–recall curve is generated for six different images. Red bounding boxes show ground truth objects while green bounding boxes indicate detected objects.

**Figure 7 sensors-23-03223-f007:**
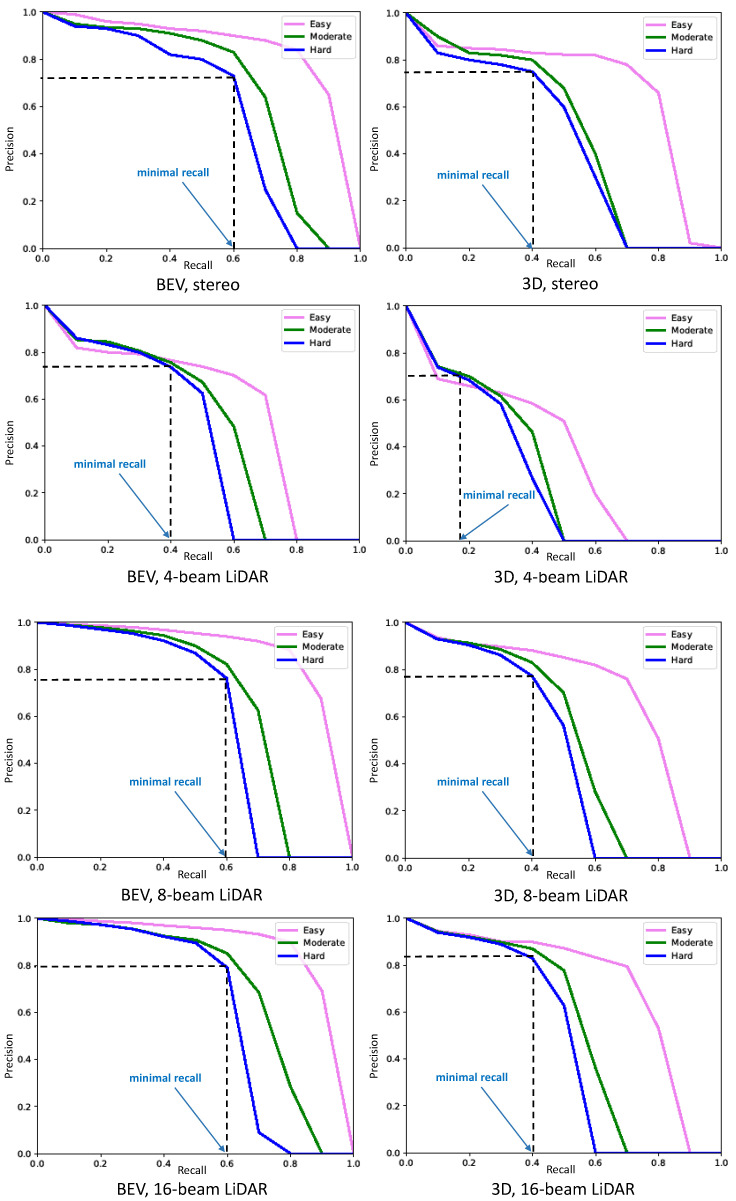
Precision–recall (P–R) curves obtained for the detection of 3D objects (right column) and 2D objects in BEV (left column). In this graph, the minimal recall shows the first recall value obtained when the P–R curve starts to drop sharply and the precision score is still higher or equal to 0.7.

**Table 2 sensors-23-03223-t002:** Comparison of some LiDAR sensors. Channels show the number of laser beams of the LiDAR sensor vertically. Range indicates the maximum distance to objects at which a LiDAR can detect. HFoV/RES and VFoV/RES decode the horizontal and vertical field of view and angular resolution, respectively. There are a number of LiDARs whose resolution depends on frequency.

Model	Channels	Range	HFoV/RES	VFoV/RES	Cost
	(Vertical)	(m)	(Degree)	(Degree)	($)
VLS-128 [43]	128	300	360°/0.2°@10 Hz	+15° to −25°/0.11°	100 k
AT128 [44]	128	200	120°/0.1°	25.4°/ 0.2°	NA
Pandar128 [45]	128	200	360°/0.1°@10 Hz	+15° to −25°/0.125°	NA
HDL-64E S2, S3 [4]	64	120	360°/0.17°@10 Hz	+2.0° to −24.9°/0.4°	75 k
Pandar64 [46]	64	200	360°/0.2°@10 Hz	+15° to −25°/0.167°	30 k
HDL-32E [47]	32	100	360°/0.2°@10 Hz	+10.67° to −30.67°/1.33°	30 k
RS-LiDAR-32 [48]	32	200	360°/0.1°	+15° to −25°/0.33°	16.8 k
VLP-16 [49]	16	100	360°/0.2°@10 Hz	±15°/2°	8 k
HS8 [50]	8	100	120°/0.18°	6.66°/0.36°	4 k
Scala [11]	4	200	145°/0.25°	3.2°/0.8°	0.6 k

**Table 3 sensors-23-03223-t003:** True and false positive-detected bounding boxes with their corresponding confidence levels. Det. and Conf. denote detection and confidence, respectively.

Images	Det.	Conf.	TP/ FP
Image 1	B1	81%	FP
Image 1	B2	71%	TP
Image 1	B3	77%	FP
Image 2	B4	67%	FP
Image 2	B5	70%	TP
Image 2	B6	87%	FP
Image 3	B7	55%	TP
Image 3	B8	65%	FP
Image 3	B9	75%	FP
Image 3	B10	85%	TP
Image 3	B11	54%	FP
Image 4	B12	57%	FP
Image 4	B13	77%	FP
Image 4	B14	59%	FP
Image 4	B15	51%	FP
Image 5	B16	61%	FP
Image 5	B17	71%	TP
Image 5	B18	91%	TP
Image 5	B19	81%	FP
Image 6	B20	88%	TP
Image 6	B21	91%	FP

**Table 4 sensors-23-03223-t004:** Precision and recall for each accumulated detection bounding box ordered by the confidence measure. Det., Conf., and Acumm. denote detection, confidence, and accumulated, respectively.

Images	Det.	Conf.	TP	FP	Accum. TP	Accum. FP	Precision	Recall
Image 5	B18	91%	1	0	1	0	1	0.09
Image 6	B21	91%	0	1	1	1	0.5	0.09
Image 6	B20	88%	1	0	2	1	0.666	0.181
Image 2	B6	87%	0	1	2	2	0.5	0.181
Image 3	B10	85%	1	0	3	2	0.6	0.272
Image 1	B1	81%	0	1	3	3	0.5	0.272
Image 5	B19	81%	0	1	3	4	0.428	0.272
Image 1	B3	77%	0	1	3	5	0.375	0.272
Image 4	B13	77%	0	1	3	6	0.333	0.272
Image 3	B9	75%	0	1	3	7	0.3	0.272
Image 1	B2	71%	1	0	4	7	0.363	0.363
Image 5	B17	71%	1	0	5	7	0.416	0.454
Image 2	B5	70%	1	0	6	7	0.461	0.545
Image 2	B4	67%	0	1	6	8	0.428	0.545
Image 3	B8	65%	0	1	6	9	0.4	0.545
Image 5	B16	61%	0	1	6	10	0.375	0.545
Image 4	B14	59%	0	1	6	11	0.353	0.545
Image 4	B12	57%	0	1	6	12	0.333	0.545
Image 3	B7	55%	1	0	7	12	0.368	0.636
Image 3	B11	54%	0	1	7	13	0.35	0.636
Image 4	B15	51%	0	1	7	14	0.333	0.636

**Table 5 sensors-23-03223-t005:** Evaluation of 3D object detection performance by a stereo camera, different types of LiDARs, and the fusion of those. In the upper part of the table, the performance (measured using the area under the curve (AUC) in the P–R curve) and the confidence interval(s) (CI) of the stereo camera and LiDAR 4, 8, 16, and 64 beams are shown with respect to three levels of difficulties for objects to detect (easy, moderate, and hard). The bottom of the table presents the detection performance and CI using fusion between a stereo camera and LiDAR, i.e., 4-beam (S+L4), 8-beam (S+L8), 16-beam (S+L16), and 64-beam (S+L64). Each result is provided according to two indicators: average precision BEV (left)/average precision 3D (right).

			Easy	Moderate	Hard
Sensors	S	AUC	**82.38/** **68.08**	**65.42/** **50.81**	**57.81/** **46.07**
		95% CI	[81.75, 82.99]/[67.31, 68.84]	[64.64, 66.19]/[49.99, 51.63]	[57.00, 58.61]/[45.26, 46.89]
	L4	AUC	56.72/38.82	49.25/32.02	44.14/29.75
		95% CI	[55.91, 57.53]/[38.03, 39.62]	[48.43, 50.07]/[31.26, 32.79]	[43.33, 44.95]/[29.01, 30.5]
	L8	AUC	84.55/68.75	65.68/50.39	58.78/45.75
		95% CI	[83.95, 85.13]/[67.99, 69.5]	[64.9, 66.45]/[49.57, 51.21]	[57.97, 59.58]/[44.94, 46.57]
	L16	AUC	85.15/70.01	68.70/52.55	60.13/47.49
		95% CI	[84.56, 85.72]/[69.26, 70.75]	[67.94, 69.45]/[51.73, 53.37]	[59.33, 60.93]/[46.67, 48.31]
	L64	AUC	**87.83/75.44**	**75.75/60.84**	**69.07/55.95**
		95% CI	[87.29, 88.35]/[74.73, 76.14]	[75.04, 76.44]/[60.04, 61.63]	[68.31, 69.82]/[55.14, 56.76]
Sensors	S+L4	AUC	**87.51/76.67**	**76.88/63.90**	**73.55/56.78**
		95% CI	[86.96, 88.04]/[75.97, 77.35]	[76.18, 77.56]/[63.11, 64.68]	[72.82, 74.26]/[55.97, 57.59]
	S+L8	AUC	87.52/76.67	76.96/63.99	73.63/56.95
		95% CI	[86.97, 88.05]/[75.97, 77.35]	[76.26, 77.64]/[63.2, 64.77]	[72.9, 74.34]/[56.14, 57.76]
	S+L16	AUC	87.74/76.88	76.98/64.10	73.91/57.05
		95% CI	[87.19, 88.27]/[76.18, 77.56]	[76.28, 77.66]/[63.31, 64.88]	[73.19, 74.62]/[56.24, 57.86]
	S+L64	AUC	**88.06/77.44**	**77.18/64.84**	**74.33/57.25**
		95% CI	[87.52, 88.58]/[76.75, 78.12]	[76.49, 77.86]/[64.06, 65.62]	[73.61, 75.04]/[56.44, 58.06]

## Data Availability

Not applicable.

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
