# Peer review of "3D Object Detection for Self-Driving Cars Using Video and LiDAR: An Ablation Study"

_sensors, 2023, doi:10.3390/s23063223_

Round 1

Reviewer 1 Report

The authors note that the contributions of this paper are an ablation study.  They specifically state "Roughly, two important results are presented: 1) an analysis of the camera stereo and LiDAR contributions in the performance of the SLS-Fusion model for 3D object detection, and 2) an analysis of the relationship between LiDAR resolution (beam number) and the accuracy achieved by the 3D object detector."

The authors' presentation of the background and technical approach are sound and do have a bit of merit to the field.  However, there are a number of issues with the document.

1. There are some grammatical issues that should be addressed.

2. Section 5.1: The level of detail of defining these metrics is unnecessary as they are well-known metrics in the field. The authors could have simply referred the reader to other papers and spent more time detailing experiments and results.

3. The following sentence in Section 5.1 is incorrectly stated, "Consider the detections as seen in Figure 5, there are 6 images with 10 ground truth objects represented by the green bounding boxes and 21 detected bounding boxes shown in red."  Ground truth images in the figure are red boxes and detections are green boxes.

4. In Section 5.2: The authors should note that there is a significant improvement by combining Stereo with a 4 beam lidar versus just using stereo alone or just using the 4 beam lidar alone.  Furthermore, the combination of stereo and the 4 beam lidar yields results that are comparable to the lidars with the higher number of beams (with and without stereo). 

5. In Section 5.2: The authors state, "...interesting to see is that increasing the number of beams does not increase detection performance".  This statement is not completely accurate.  It does increase detection performance, but the question is how significant?  This can (and should) be quantified by doing a statistical analysis.

6. Section 5.2:  There are many publicly available datasets.  The authors should have explored those datasets rather than simply state that KITTI may be ineffective.

Reviewer 2 Report

The manuscript verified a cost-effective SLS-fusion model through ablation study, which combines the Lidar and stereo camera. The impact of each component is investigated by increasing the number of beams of Lidar, which results in rigorous demonstration. What have to be aware of is that some problems as following need to be resolved and answered yet in the manuscript. 

1.      In your manuscript, it is supposed that the number of Lidar beams equals to resolution. Actually, high resolution means that the divergence angle between probed beams is small. Therefore, increasing the number of beams will not definitely improve the resolution. Such a precondition in this work needs to be clearer. Otherwise, it is confused that the results are caused by ‘dense’ or ‘numerous’ point clouds.

2.      In Table 1, several methods are compared on three levels of difficulty: Easy, Moderate and Hard. In the following sections, this difficulty grading occurs as part of experiments. So, how to define the levels of difficulty? The serious definition helps readers to follow this work.

3.      As the Table 1 shows, the fusion solution performs obviously worse than CG-stereo and PointRCNN on easy level. The reason behind should be explained so that the advantage of fusion model could be convinced.

4.      For experiments in this work, is it possible to show a real scene that is detected or imaged? The results may be understanded directly from the comparison between the real scene and detected results.

5.      How to find the minimal recall exactly? What does the minimal recall represent physically?

Reviewer 3 Report

This manuscript sensors-2224907 investigates analyze how the stereo and LiDAR sensors contribute to the performance of the SLS-fusion model for 3D object detection. It is known that data coming from the stereo camera has a significant role in the fusion model. However, it is necessary to quantify this contribution, as well as to identify the variation of such contribution with respect to the number of LiDAR beams used inside the model. So, to evaluate the role of the component parts of the SLS-fusion network that represent LiDAR and stereo camera architectures, we propose to divide the model into two independent decoder networks. The results of this study show that starting from 4 beams, increasing the number of LiDAR beams has no significant impact on the SLS-fusion performance. The presented results can guide design decisions by practitioners. I feel the experiment results are sufficient. It was a pleasure reviewing this work and I can recommend it for publication in Sensors after a major revision. I respectfully refer the authors to my comments below.

1.         The English needs to be revised throughout. The authors should pay attention to the spelling and grammar throughout this work. I would only respectfully recommend that the authors perform this revision or seek the help of someone who can aid the authors.

2.         (References) Please adjust the style of all the references to meet the Sensors Journal requirement.

3.         (Page 15) The original figures 4 and 7 is not clear. Please redraw this figure clearly. Add some word in this figure, and indicate the imaging modules.

4.         (Section I, Introduction) The reviewer suggest to revise the original statement as “Generally, a CNN (Convolutional Neural Network) is then used to take advantage of this information <*>. ("GMDL: Toward precise head pose estimation via Gaussian mixed distribution learning for students’ attention understanding," Infrared Physics & Technology, 2022.; "NGDNet: Nonuniform Gaussian-label distribution learning for infrared head pose estimation and on-task behavior understanding in the classroom," Neurocomputing, 2021.)

5.         (Section 1 Introduction) The reviewer hopes the introduction section in this paper can introduce more studies in recent years. The reviewer suggests authors don't list a lot of related tasks directly. It is better to select some representative and related literature or models to introduce with certain logic. For example, the latter model is an improvement on one aspect of the former model.

6.         The reviewer suggests to add a new paragraph in Introduction part to summary the contribution of this manuscript.

7.         (Page 6) The original statement “In fact, the study of the structure of neural networks has become best practice for machine learning research <*>, …”. (Facial expression recognition method with multi-label distribution learning for non-verbal behavior understanding in the classroom. Infrared Physics & Technology 2021, 112, 103594.;; Learning fusion feature representation for garbage image classification model in human–robot interaction. Infrared Physics & Technology 2023, 128, 104457.)

8.         Experimental pictures or tables should be described and the results should be analyzed in the picture description so that readers can clearly know the meaning without looking at the body.

9.         (Table 1) All the values in this table should be with same data accuracy. The number of data after the decimal point are the same. Please check other Tables.

10.     The authors are suggested to add some experiments with the methods proposed in other literatures, then compare these results with yours, rather than just comparing the methods proposed by yourself on different models.

11.     Discuss the pros and cons of the proposed model.

My overall impression of this manuscript is that it is in general well-organized. The work seems interesting and the technical contributions are solid. I would like to check the revised manuscript again.

Round 2

Reviewer 1 Report

The authors reasonably addressed each of the reviewer's previous issues.  The research could still be improved if the authors conducted the study on other datasets and specifically discussed the results with respect to each dataset's specific data.  Also, there are still some minor grammatical errors that should be addressed.

Author Response

Thanks again for your invaluable work. Next, you will find our answers to your comments.

Reviewer 1:

The authors reasonably addressed each of the reviewer's previous issues.  The research could still be improved if the authors conducted the study on other datasets and specifically discussed the results with respect to each dataset's specific data.  Also, there are still some minor grammatical errors that should be addressed.

Remark:

“The research could still be improved if the authors conducted the study on other datasets and specifically discussed the results with respect to each dataset's specific data.”

Answer:

  • Of course, we plan to deepen the assessment by using other datasets and also real data coming from an autonomous vehicle. This is already mentioned in section 5.2 Ablation results (line 482) and section 6. Conclusion (lines 516-520: "However, to better generalise the model and to find the optimal solution between the performance of detecting obstacles and the price of the LiDAR component, it is desirable to test the model in many different datasets and environments such as waymo [55 ],nuScenes [56] or argoverse2 [57] datasets. "). Considering the current publishing timescales and the relative sparsity of suitable public datasets, we look forward to presenting further results in a follow-on paper.

Remark:

“Also, there are still some minor grammatical errors that should be addressed.”

Answer:

  • Thank you for your comment on grammatical issues. We have now double-checked the manuscript through a bilingual speaker. Grammatical corrections are reported in yellow in the pdf document.

Reviewer 2 Report

No futhre comments on the revised version.

Author Response

Thanks again for your invaluable work.